# Effect of Selected Bio-Components on the Cell Structure and Properties of Rigid Polyurethane Foams

**DOI:** 10.3390/polym15183660

**Published:** 2023-09-05

**Authors:** Aleksander Prociak, Michał Kucała, Maria Kurańska, Mateusz Barczewski

**Affiliations:** 1Department of Polymer Chemistry and Technology, Faculty of Chemical Engineering and Technology, Tadeusz Kosciuszko Cracow University of Technology, Warszawska 24, 31-155 Krakow, Poland; maria.kuranska@pk.edu.pl; 2Institute of Materials Technology, Poznan University of Technology, Piotrowo 3, 61-138 Poznan, Poland; mateusz.barczewski@put.poznan.pl

**Keywords:** rigid polyurethane foams, bio-polyols, renewable resources, mechanical properties, physical properties

## Abstract

New rigid polyurethane foams (RPURFs) modified with two types of bio-polyols based on rapeseed oil were elaborated and characterized. The effect of the bio-polyols with different functionality, synthesized by the epoxidation and oxirane ring-opening method, on the cell structure and selected properties of modified foams was evaluated. As oxirane ring-opening agents, 1-hexanol and 1.6-hexanediol were used to obtain bio-polyols with different functionality and hydroxyl numbers. Bio-polyols in different ratios were used to modify the polyurethane (PUR) composition, replacing 40 wt.% petrochemical polyol. The mass ratio of the used bio-polyols (1:0, 3:1, 1:1, 1:3, 0:1) affected the course of the foaming process of the PUR composition as well as the cellular structure and the physical and mechanical properties of the obtained foams. In general, the modification of the reference PUR system with the applied bio-polyols improved the cellular structure of the foam, reducing the size of the cells. Replacing the petrochemical polyol with the bio-polyols did not cause major differences in the apparent density (40–43 kg/m^3^), closed-cell content (87–89%), thermal conductivity (25–26 mW⋅(m⋅K)^−1^), brittleness (4.7–7.5%), or dimensional stability (<0.7%) of RPURFs. The compressive strength at 10% deformation was in the range of 190–260 and 120–190 kPa, respectively, for directions parallel and perpendicular to the direction of foam growth. DMA analysis confirmed that an increase in the bio-polyol of low functionality in the bio-polyol mixture reduced the compressive strength of the modified foams.

## 1. Introduction

Today, the transition to a climate-neutral economy is an urgent matter and an opportunity for a better future. The balance between greenhouse gas emissions and their storage and absorption by waters, forests, and soil is one of the most important challenges related to climate protection. The actions aimed at a climate-neutral economy include also replacing petrochemical raw materials with renewable ones [1].

Polyurethanes (PURs) are an important group of polymeric materials used in many industry sectors. A balanced approach to the synthesis of PURs can be implemented in different ways. One of them is replacing polyols (one of the two main ingredients in the synthesis of PURs) with bio-polyols obtained from renewable raw materials. The other method to obtain more sustainable PUR materials is the modification of the PUR matrix with natural fillers [2,3,4]. Bio-polyols can be synthesized using different sources, i.e., from vegetable oils and waste materials in accordance with the idea of a circular economy and following the green chemistry principles [5,6]. Most of the PURs produced in industry are rigid and flexible foams. Rigid PUR foams are the most effective thermal insulation materials, which can be characterized by closed- and open-cell structures.

Rigid PUR foams are primarily used as thermal insulation materials, and therefore, a low thermal conductivity coefficient (λ) is one of their most important properties. The mechanism of heat transfer in porous materials is complex. Generally, it is a result of gas conduction of closed foam cells, conduction of solid polyurethane, radiation, and convection. Due to the very small size of the cells in rigid foams, convection is often omitted [7,8,9]. The temperature of thermal conductivity measurement has influence on its value. Zhang et al. measured the thermal conductivity of PUR foams obtained with different blowing agents using the transient plane source (TPS) method at different temperature and humidity. The thermal conductivity of the five PUR foams was measured from −40 °C to 70 °C in the high- and low-temperature test chamber with an interval of 5 °C. Thermal conductivity, regardless of the blowing agent used to prepare the foams, increases non-monotonicity with temperature, because the increase in temperature favors the diffusion of gases in closed cells and thermal conductivity, and the phases of the components in closed cells also change with temperature [10]. Thermal conductivity measurements are difficult to perform, especially for temperatures above the glass transition temperature. The high deformability of the material as the temperature increases prevents proper contact of the sensor with the material surface. Additionally, such measurements are not possible because the material decomposes. However, Duarte et al. developed a numerical procedure capable of estimating the temperature dependent effective thermal conductivities and specific heats of PUR foams for temperatures well above those corresponding to their decomposition [11].

Rigid PUR foams can be used as thermal insulation materials in the building industry as well as in liquefied natural gas (LNG) tanks because of its superior mechanical and thermal insulating properties. Taking into account the various applications of rigid PUR foams, their mechanical properties are very significant. Byeong-Kwan et al. investigated the effects of foam apparent density, temperature, and impact energy on the dynamic behavior of PUR foams. Dynamic compression tests were conducted on rigid PUR foams characterized by different apparent densities (90–180 kg/m^3^) at various temperatures (25, −70, and −163 °C). It was concluded that with an increase in apparent density and a decrease in temperature, the foams’ mechanical properties increased [12]. Linul et al. also tested closed-cell rigid PUR foams of different densities (100, 160, and 300 kg/m^3^). The specimens were subjected to uniaxial dynamic compression with a loading rate in the range of 1.37–3.25 m/s with different temperatures (20, 60, 90, and 110 °C). Experimental results showed that Young’s modulus, yield stress, and plateau stress values increased with increasing foam apparent density. They concluded that the most significant parameter of mechanical properties in dynamic compression of rigid PUR foams is the foam apparent density; however, the temperature of the test, loading rate, and material orientation also have to be taken into account [13].

The thermal properties of polyurethanes, which determine the scope of their applications, are described by two characteristic temperature values: the degradation temperature (T_D_) and the glass transition temperature (T_g_). The first one is determined using thermogravimetric analysis (TGA). Depending on the substrates used, i.e., the type of isocyanate and types of polyol [14], which translates into the content of rigid and soft segments in the foam, the onset of the decomposition temperature determined at 5% weight loss varies in the reported literature in the range of 200–296 °C [15,16]. Along with the increasing concentration of the addition of substances of natural origin, including bio-polyols, the degradation temperature of the PURs decreases [14].

The second characteristic temperature, the glass transition temperature, is associated with a change in the material’s thermomechanical properties. It is defined by Hutchinson [17], based on Rehage and Borchard [18], as follows: “*The glass transition represents the change that occurs when a system, initially in an equilibrium liquid-like or rubbery state defined by the thermodynamic variables temperature (T) and pressure (P) (and possibly also by other variables, such as composition in chemically reacting systems), transforms into a non-equilibrium glassy state as a result of a restriction of the molecular mobility, corresponding to an increase in the average relaxation time*”. Determination of T_g_ can be carried out using various methods, but the most commonly used methods are dynamic thermomechanical analysis (DMA) and differential scanning calorimetry (DSC) [17,19,20]. Depending on the measurement method, the values of the T_g_ of rigid foams determined by the DMA method, which contains information about the material’s cellular structure, are described in the literature in a wide range from approx. 100 to even 200 °C [21,22,23,24]. Lower values were recorded, e.g., by Tan et al. [25] for rigid polyurethane foams containing soybean oil-based polyol in a system with polymeric methylene diphenyl diisocyanate (PMDI). Ye et al. determined, based on the change in loss modulus, the glass transition temperature for PMDI-based flame-retarded high-density rigid PUR foams modified with expanded graphite T_g_ at the level of 176–191 °C [21]. It should be emphasized that the differences in T_g_ are mostly connected to cross-link density and type of reagents, including isocyanates. Among the commonly used isocyanates, the highest glass transition temperatures are noted for systems based on aromatic isocyanates, particularly PMDI [16].

More and more scientists are developing PUR systems for thermal insulation purposes based on bio-polyols from vegetable oils. One of the most interesting methods for bio-polyol synthesis is the epoxidation of triglyceride bonds and subsequent ring-opening with nucleophiles [26,27]. The possibility of using various nucleophilic compounds offers a variety of chemical structures of the resultant bio-polyols. Bio-polyols synthesized by epoxidation and opening oxirane rings with monohydric alcohols are characterized by the presence of secondary hydroxyl groups only, the reactivity of which with isocyanate groups is lower than the reactivity of primary groups. In the case of oxirane ring agents having two hydroxyl groups, it is possible to synthesize bio-polyols with primary and secondary hydroxyl groups [28]. The type of oxirane ring-opening agent affects the hydroxyl value of bio-polyols and their chemical structures. Uram et al. synthesized bio-polyols based on rapeseed oil using epoxidation and the oxirane ring-opening method [28]. In their work, two types of oxirane rings agents (1-hexanol and 1,6-hexanediol) were used in order to obtain bio-polyols with different functionalities as well as containing primary and secondary hydroxyl groups. The hydroxyl value of the bio-polyol with 1,6-hexanediol (250 mgKOH/g) was more than twice as high as that of the bio-polyol obtained with 1-hexanol (104 mgKOH/g). They concluded that it is possible to produce good-quality rigid foams by replacing the petrochemical polyol in a PUR system with such bio-polyols up to 40%. However, the chemical structures of the bio-polyols had an influence on their properties. The modification of the PUR system with the bio-polyol of a lower hydroxyl value caused a decrease in the compressive strength due to a lower cross-linking density of the modified foams. In the case of the PUR system modified with the higher-functionality bio-polyol, the system reactivity was higher than in the case of other system with the bio-polyol of lower functionality. The advantage of the lower-functionality bio-polyol was its lower viscosity (643 mPa·s) compared to that of the higher-functionality bio-polyol (5128 mPa·s). The viscosity of polyols is a crucial parameter, as it has a major impact on the mixing of PUR formulations. Reducing the viscosity of PUR systems is an important issue from a processing point of view [29].

Prociak et al. [30] reported on rigid PUR foams prepared using different contents of a mixture of two bio-polyols (20–40 wt.% in polyol mixture) with the same characteristics as those applied by Uram et al. [28]. They found that biofoams with apparent densities comparable to the material based on petrochemical polyols were characterized by similar values of thermal conductivity and a decrease in mechanical strength. Despite the deterioration of the mechanical properties caused by the plasticization of the PUR matrix with bio-polyols containing dangling chains, all materials were dimensionally stable at room temperature.

It is therefore reasonable to undertake research using bio-polyols with different properties, such as hydroxyl number, type of hydroxyl groups, functionality, and viscosity, mixed in different ratios (1:0, 3:1, 1:1, 1:3, 0:1), in order to study the influence of such mixtures on the foaming process and selected properties of PUR biofoams.

## 2. Materials and Methods

### 2.1. Materials

In order to prepare rigid polyurethane foams (RPURFs), petrochemical polyether polyol (Rokopol^®^ RF 551 from PCC Rokita SA, Brzeg Dolny, Poland) and two bio-polyols obtained as a result of rapeseed oil epoxidation and opening oxirane rings of the epoxidized oil were used. In the synthesis of the bio-polyols, 1-hexanol and 1,6-hexanediol were used as hydroxylating reagents. The symbols of the obtained bio-polyols come from the type of oxirane ring openers. The bio-polyols were previously synthesized in the Department of Chemistry and Technology of Polymers of the Faculty of Chemical Engineering and Technology of the Cracow University of Technology. Figure 1 shows hypothetical chemical structures of the C.HEX and C1.6HEX bio-polyols.

The characteristics of the polyols used to obtain RPURFs are presented in Table 1. In the conducted tests, 40% by mass of the petrochemical polyol was replaced with the bio-polyols in following mass ratios of C.HEX:C.1.6HEX: 1:0, 3:1, 1:1, 1:3, 0:1.

The second raw material, necessary in the production of PUR materials, used in the synthesis reaction of the tested RPURFs was polymeric diphenylmethane diisocyanate (PMDI) from Minova Ekochem SA (Siemianowice Śląskie, Poland). The used isocyanate contained 31 wt.% of free isocyanate groups. Moreover, a catalyst was also used, which is a chemical compound belonging to the group of tertiary amines, and an organosilicon surfactant. The process of foaming the PUR compositions was carried out using a chemical blowing agent, which was demineralized water. The reaction of demineralized water with isocyanate components leads to the generation of carbon dioxide, which expands the reaction mixture.

Table 2 shows the formulations of the tested RPURFs. The mass shares of the components were calculated in relation to 100 g of the polyol mixture used in the composition. The names of the samples come from the mass percentage of the bio-polyols in the reaction mixture. C.HEX bio-polyol is designated as P1 and C.1.6HEX bio-polyol as P2. For example, the sample with the name P1/P2_75/25 contained C.HEX and C.1.6HEX bio-polyols in a mass ratio of 3:1.

### 2.2. Synthesis of RPURFs

RPURFs based on the petrochemical polyol and the bio-polyols were obtained in a single-stage process. At the beginning of the synthesis, a so-called polyol premix was obtained, containing polyol components, catalyst, surfactant, and water. The components forming the polyol premix were mixed for 60 s in the appropriate weight proportions, according to a given formulation. In the next step, the isocyanate component was added to the polyol premix and mixed for 5 s. The obtained PUR compositions were poured into molds (25 cm × 25 cm × 10 cm) in which the RPURFs were allowed to rise freely. The obtained foams were seasoned at 21°C for 24 h.

### 2.3. Research Methodology

The cellular structure and selected physical and mechanical properties of the prepared RPURFs were tested.

The study of the cellular structure of RPURFs was carried out using an optical microscope (PZO, Warsaw, Poland). Thin monolayers of RPURFs were cut out using a scalpel in the cross-sections parallel and perpendicular to the direction of foam growth. Five microscopic images of each foam structure were taken. Using the Aphelion^TM^ software (version 3.1), the obtained images were analyzed, and the average cross-sectional area of the cells and the average height and width of the pores were determined. The average height and width of the cells allowed for the calculation of the cell anisotropy index.

In order to evaluate the closed-cell content in RPURFs, four samples of approximately 2.5 cm × 2.5 cm × 10 cm were cut from each type of obtained foam material. The exact dimensions of the samples were then determined using a caliper. The tests were carried out according to the PN-ISO 4590 standard [32].

Apparent density measurements were carried out in accordance with the ISO 845 standard [33]. Shapes of approximately 20 cm × 20 cm × 5 cm were cut out of the foam material. Using a caliper, the exact values of the length, width, and height of the samples were determined. The measured values made it possible to calculate the volume of the foam material. After weighing the samples, the apparent density of the RPURFs was calculated as the ratio of the sample mass to its volume.

In order to determine the thermal conductivity coefficient of the RPURFs, samples of approximately 20 cm × 20 cm × 5 cm were cut out. The samples were placed in a Laser Comp Heat Flow Instrument Fox 200 thermal conductivity apparatus between two plates (TA Instruments, New Castle, DE, USA). The lower plate kept the temperature at 20 °C and the upper plate at 0 °C. The apparatus for the thermal conductivity coefficient measurement was made in accordance with the ISO 8301 standard [34].

The compression tests were carried out using a ZwickRoell testing machine model Z005 TH Allround-Line, and the foam samples in the cylinder shape were about 4 cm in diameter and height. The exact dimensions of the rolls were entered into the computer program controlling the testing machine. Then, the cylinders were compressed to 10% of their original height. After the test, the stress value was read at 10% compression of the sample. Compressive strength tests of the foams were carried out in accordance with the PN-EN ISO 844 standard [35] in parallel and perpendicular directions to the direction of foam growth.

Twelve cubes of approximately 2.5 cm × 2.5 cm × 2.5 cm were cut from the RPURFs in order to conduct brittleness tests. The samples were weighed and placed in a brittleness apparatus. This device is an oak chest with 24 oak cubes measuring 2 cm × 2 cm × 2 cm. After closing the box, together with the oak cubes and the tested samples, the device was turned on and rotated on its axis for 10 min. After 10 min, the device was turned off and, after thorough cleaning, the foam cubes were weighed again. The difference in the mass of the foam cubes before and after the test determined the mass loss of the samples. The brittleness measurements were carried out according to ASTM C-421-61 [36].

The dynamic mechanical properties of the foams were analyzed using a rotational rheometer Anton Paar MCR 301 (Graz, Austria) equipped with an SRF accessory for dynamic mechanical thermal analysis (DMTA). Samples with dimensions of 10 mm × 10 mm × 50 mm were tested with the torsion mode. The tests were carried out at a heating rate of 2 °C/min at the temperature range of 30 °C to 220 °C. The applied frequency and strain were 1 Hz and 0.01%, respectively. The storage modulus (G′) and damping factor (tan δ) were determined.

In order to evaluate the dimensional stability of the tested materials, four samples of 10 cm × 10 cm × 2.5 cm were cut from each RPURF. Using a caliper, the length and width of the samples (at three specific locations) and the height of the samples (at four specific locations) were measured. The samples were kept for 24 h at −25 °C, as well as at 70 °C and 90% humidity. After removing the samples from the freezer and climatic chamber, they were kept for 1 h at room temperature before measuring them. From the obtained measurement data, differences in linear dimensions of the samples were calculated. Dimensional stability tests were carried out in accordance with the ISO 2796-1986 standard [37].

## 3. Results and Discussion

### 3.1. Cell Structure

The cellular structure of RPURFs depends on the initial viscosity of the reaction system and the foaming process, the course of which depends on the foaming and gelation reactions.

The modification of RPURFs with the bio-polyols caused a slight increase in the cell anisotropy index in the cross-section parallel to the foam growth (Table 3).

The increase in the anisotropy index of the RPURF cells in the parallel cross-section of the P1/P2_75/25 foam increased by a maximum of about 8% in relation to the value of the anisotropy index of the reference sample. The P1/P2_75/25 foam had the highest anisotropy index in this cross-section.

With the increase in the share of P2 in the foam material, the cell surface area also increased in the cross-section parallel to the foam growth direction. On the other hand, the P1/P2_0/100 foam had the highest average value in the cross-sectional cell area, reaching a value 5% greater than in the case of the reference foam.

Considering the cross-section of the foam in the direction perpendicular to its growth, it was noticed that the anisotropy index of the cells was slightly lower (maximum by about 4%) or equal to the anisotropy index of the cells in the reference material. The P1/P2_50/50 and P1/P2_0/100 samples had cell anisotropy index values equal to that of the reference sample.

The P1/P2_50/50 sample had the smallest values of average cell surface area in the cross-section perpendicular to the direction of foam growth, reaching a value about 4% lower than in the case of the reference material. The cell surface area values in the perpendicular cross-section for all foams modified with the bio-polyols were lower than the corresponding values of the reference material.

Comparing the results of the anisotropy index and the cross-sectional area of the cells, the tendency of the anisotropy index to increase with the decrease in the cross-sectional area of the cells was noticed. When analyzing the cross-sections of the tested foams in the parallel direction, it was noticed that the values of the anisotropy index and the cross-sectional area of the cells were greater than in the case of the cross-section perpendicular to the direction of growth of those foams.

The value of the anisotropy index of greater than 1 in the case of the parallel cross-section of the RPURFs and of smaller than 1 in the case of the perpendicular cross-section was caused by free growth of the foams in a mold and an elongation of cells in the foam growth direction. After analyzing the results, it was found that the modification of the RPURFs with the bio-polyols did not significantly affect the value of the anisotropy index of the modified foams.

Table 4 presents exemplary photographs of the cellular structure of the tested RPURFs in the cross-sections both parallel and perpendicular to the direction of foam growth.

Based on the reviewed literature, it was found that in the case of previously studied foams with 40% of the bio-polyols in a polyol mixture, the results of the anisotropy index and the cross-sectional area of cells were similar [28]. A slight difference may have been a result of various types of components used in the foam formulations.

The percentage of closed cells in the RPURFs modified with the bio-polyols did not differ significantly from the value for the reference material (Figure 2). All modified foams had a lower percentage of closed cells than the unmodified foam. The lowest content of closed cells was found for the P1/P2_100/0 sample, which was about 6% less than in the case of the reference foam. Taking into account the standard deviations, it was concluded that the tested foams were characterized by similar values of closed-cell content compared with the materials described in the literature. The literature [30] shows that replacing 40% by mass of petrochemical polyol with bio-polyol resulted in obtaining foam materials with a closed-cell content of about 89%.

### 3.2. Apparent Density and Thermal Conductivity Coefficient

The apparent density of RPURFs depends on many factors, including component reactivity, the type and amount of blowing agents, and catalysts that significantly influence the foaming process of reaction mixtures. Apparent density values allow for types of PUR products to be distinguished. RPURFs generally have an apparent density in the range of 30 to 60 kg/m^3^.

The thermal conductivity coefficient is the main parameter determining the usefulness of RPURFs as thermal insulation. The lower the thermal conductivity coefficient, the better the thermal insulation properties of a given material.

The RPURFs modified with the bio-polyols were characterized by similar apparent density values compared to the reference foam (Table 5). The P1/P2_75/25 foam had the largest apparent density deviation from the reference foam, of approximately 5%. With the increase in the share of P2 bio-polyol in the foam material, the value of its apparent density increased. The P1/P2_0/100 sample was the only one with a density value higher (by about 2%) than the reference material. For samples P1/P2_100/0 and P1/P2_0/100, the apparent density values were higher by about 9% and 11%, respectively, compared to the information given in the literature [28]. Sample P1/P2_50/50 had an apparent density higher by about 7% compared to the foam with the same mass fraction of bio-polyols given in the literature [31].

The modification of the RPURFs with the bio-polyols did not significantly affect the values of the thermal conductivity coefficient of foams in relation to unmodified materials. The difference in value was a maximum of about 4%. The thermal conductivity of all modified foams, except for the P1/P2_75/25 sample, had higher values than the reference material. This proves slightly worse thermal insulation properties of the foams synthesized with the bio-polyols in comparison to the reference foam. The P1/P2_75/25 foam had the closest value of the thermal conductivity coefficient to the reference material. The values of the thermal conductivity coefficient of P1/P2_50/50, P1/P2_100/0, and P1/P2_0/100 foams modified with the bio-polyols were similar to values reported in the literature [28,31].

The anisotropy index increased as an effect of decreasing the apparent density of the RPURFs (Figure 3). This relationship occurred in the case of the anisotropy index of foam cells evaluated for a cross-section parallel to the direction of foam growth. This was connected to the increase in cell size.

Increasing the percentage of closed cells in the foam material reduced the value of the thermal conductivity coefficient, giving the foam materials better thermal insulation properties (Figure 4). This is due to the fact that approximately 50% of the total heat transfer in a porous material takes place in the form of conduction through the gas enclosed in the cells of RPURFs [38]. Therefore, the aim is to obtain foam materials containing more than 90% of closed cells in their structure.

### 3.3. Mechanical Properties

The mechanical properties of RPURFs depend on the structure of the PUR matrix, the cell structure, and the apparent density of the foams. The values of stress at 10% compression were lower in the perpendicular direction than in the direction parallel to the foam growth direction (Figure 5). The character of the stress–strain curves is typical of rigid polyurethane foams. Differences in the maximum value of compressive stress are the effect of changes in the so-called elastic area [10], which for the tested foams was observed at a deformation of up to 7%. Generally, rigid PUR foams show linear elasticity at low stresses followed by a long collapse plateau. In the case of closed-cell foams, this linear elasticity is related to the cell wall bending and the cell face stretching [39]. Both these behaviors, in the case of the tested foams, depended on the share of bio-polyols used in the foam formulation. The different post-yield softening among the specimens tested was especially related to the content of the bio-polyol mixture. The highest plasticization effect was observed for the foams modified with the low-functionality bio-polyol (sample P1/P2_100/0) only. This effect was related to a lower crosslinking density of solid PUR as well as a content of dangling chains from the C.Hex bio-polyol (Figure 1).

The maximum compressive stress values correlated with the content of high-functionality bio-polyol in the bio-polyol P1/P2 mixture. Increasing the mass fraction of P2 bio-polyol in the mixture of polyols caused an increase in stress at 10% compression measured in both the parallel and perpendicular directions (Table 6). Sample P1/P2_0/100 had mechanical properties similar to those of the reference foam. All tested materials had lower stress values at 10% compression in the perpendicular direction compared to the value of this parameter for the reference foam. The use of bio-polyols as one of the components in the synthesis of RPURFs caused the plasticization of the PUR matrix and thus the deterioration of the mechanical parameters of the foams. Better mechanical properties of RPURFs in a parallel direction were obtained by modifying the foams with P2 bio-polyol, which was characterized by higher functionality than P1 bio-polyol.

Stress values at 10% compression in the directions parallel and perpendicular to the direction of material growth became higher with the increase in the foam apparent density (Figure 6). In the RPURFs, most of the PUR material was located in the foam’s structure parts, the so-called struts, which are connections of more than two cells. The increase in foam apparent density led to an improvement in its mechanical properties.

The brittleness of the tested foams was generally similar. However, it was found that it decreased with the content of P2 bio-polyol in the mixture of the used bio-polyols (Figure 7). The P1/P2_0/100 foam was characterized by the best mechanical properties among the tested foams and was slightly better compared to the reference material.

Similar relationships regarding the differences in stress at 10% compression in directions parallel and perpendicular to the direction of foam growth and the differences in brittleness depending on the mass fraction of selected bio-polyols have been described in the literature [28,30,31].

### 3.4. Thermomechanical Properties

Hagen et al. [40] demonstrated that the measured values of materials assessed by various dynamic thermomechanical methods may differ. This is due to different geometries of the measuring systems or heat transfer conditions. Despite the differences in the values of the characteristic relaxation temperatures or the modules, the changes and trends caused by the influence of different reagents remain consistent, and the shifts in the characteristic thermomechanical parameters can be clearly compared. The curves in Figure 8 show changes in the storage modulus and damping factor depending on the temperature. As the temperature increased, the storage modulus decreased. It was observed that the foam that contained only P2 bio-polyol had the lowest value of the storage modulus in the whole range of tested temperatures. All modified foams tested below approximately 180 °C had a lower storage modulus value than the reference foam. Above 180 °C, the sample containing only P1 bio-polyol had the highest storage modulus value.

In the temperature range of 30 °C to about 190 °C, the damping factor of the foams increased. Above the mentioned temperature, the foam-damping factor decreased. The highest damping factor values of the tested samples occurred in the temperature range of about 180 °C to about 200 °C. The foam modified only with P1 bio-polyol had the highest damping factor value, at a temperature of about 200 °C. The foam modified only by P2 bio-polyol had the lowest value for the damping factor, in the range of 180 °C to 200 °C. The foams with bio-polyol mass fractions of 25:75 and 50:50 had a damping factor value closest to the reference value.

Upon analyzing the course of curves G’(T), one can observe a gradual decrease in foam stiffness with an increasing share of P2 bio-polyol with respect to P1 bio-polyol. This effect is related to both the change in the cross-linked PUR structure and the geometry of the cells. The growing share of P2 bio-polyol in the reactive composition increased the average cell cross-section area, which decreased the porous material’s stiffness. At the same time, despite the lower functionality of P1 bio-polyol, the maximum temperature of the peak on the tanδ for samples with a dominant content of this bio-polyol was higher, which suggests a higher cross-link density of PUR [41]. The measured values of the rigid segment relaxation temperature, determined as the maximum damping factor with the corresponding value of tanδ, were REF (191 °C; 0.334), P1/P2_100/0 (200 °C; 0.374), P1/P2_75/25 (198 °C; 0.317), P1/P2_50/50 (195 °C; 0.328), P1/P2_25/75 (189 °C; 0.33), and P1/P2_0/100 (189 °C; 0.300). Similar values for the maximum damping factor, interpreted as the glass transition temperature (Tg), in the range of 194–197 °C, were noted by Fang et al. [42], who produced rigid polyurethane foams using soy-based polyols with a hydroxyl value comparable to that obtained for P2 bio-polyol. Despite different measuring conditions, compression mode, and heating rate of 10 °C/min, the relaxation temperature of rigid segments in the range of about 200 °C was also noted by Zhang et al. [23] for rigid polyurethanes based on oilseed rape straw-based polyol. Postponing to higher temperature values of a maximum of the tanδ(T) curve in the relaxation range attributed to the rigid PUR segments can be associated more with the lower mobility of macromolecules and not the geometrical structure of the foam. On the other hand, with a more significant share of P1 bio-polyol, gradually increasing tanδ at peak values was noted, which is related to the better damping ability and may confirm the plasticizing effect of this bio-polyol. The addition of a polyol with higher functionality and a comparable hydroxyl value, as demonstrated by Ionescu et al. [43], increased the glass transition temperature of castor oil-based polyurethanes, regardless of the type of isocyanate used. In the considered case, the systems containing higher concentrations of P1 bio-polyol with significantly lower viscosity may have been more miscible with a petrochemical polyol, which may directly affect the polyurethane curing process.

To conclude, the DMA analysis concerning the results of mechanical tests and analyses of the cellular structure allows us to assume that although P1 bio-polyol reduced the compressive strength and the stiffness of the foam material structure, a wide temperature range would be more favorable for material series with lower P2 bio-polyol content, which is associated with the dominant impact of cells with smaller sizes.

### 3.5. Dimensional Stability

Seasoning of RPURFs for 24 h at −25 °C as well as at 70 °C and 90% humidity did not cause significant changes in the linear dimensions of the foams (Table 7). The linear dimensions of the samples changed by a maximum of about 0.7%.

The foams modified with the bio-polyols, similarly to foams obtained using only the petrochemical polyol, were characterized by high dimensional stability. The change in the linear dimensions of the foams in different conditions did not exceed 1%.

## 4. Conclusions

Bio-polyols derived from rapeseed oil are a promising, environmentally friendly alternative to petrochemical polyols for the production of RPURFs. The type of oxirane ring used in the bio-polyol synthesis has a significant impact on its characteristics. The use of 1-hexanediol made it possible to obtain a bio-polyol with a low viscosity (< 600 mPa⋅s) and a functionality of 2.5, whereas a bio-polyol with a viscosity of about 2000 mPa⋅s and a functionality of 4.4 was obtained using 1,6-hexanediol.

In general, the modification of the reference PUR system with the applied bio-polyols improved the cellular structure of the foam, reducing the size of the cells. On the other hand, the share of the bio-polyols used in the PUR system had an impact on the cellular structure of the modified foams. The increase in the share of low-functionality bio-polyol in the bio-polyol mixture caused a decrease in the average cell surface area by about 13% (compared to the reference foam) in the cross-section parallel to the direction of foam growth. The cell anisotropy coefficient in the cross-sections parallel and perpendicular to the direction of foam growth showed slight differences in relation to the reference sample.

Replacing the petrochemical polyol with the bio-polyols did not cause major differences in apparent density, closed-cell content, thermal conductivity, or dimensional stability of RPURFs. This allowed the used petrochemical polyol to be replaced with bio-polyols and good-quality foams with a thermal conductivity of about 25–26 mW⋅(m⋅K)^−1^ to be obtained.

Regardless of the composition of the bio-polyol mixture, the modified foams were characterized by very good dimensional stability at low and high temperatures. Changes in linear dimensions were less than 0.7%, which ensured good mechanical properties of the foams. The compressive strength at 10% deformation was in the range of 190–260 and 120–190 kPa, respectively, for directions parallel and perpendicular to the direction of foam growth. The compressive stresses depend mainly on the apparent density of the foam. However, increasing the proportion of high-functionality bio-polyol in the bio-polyol mixture had a beneficial effect on both the compressive strength and the brittleness of the foams.

## Figures and Tables

**Figure 1 polymers-15-03660-f001:**
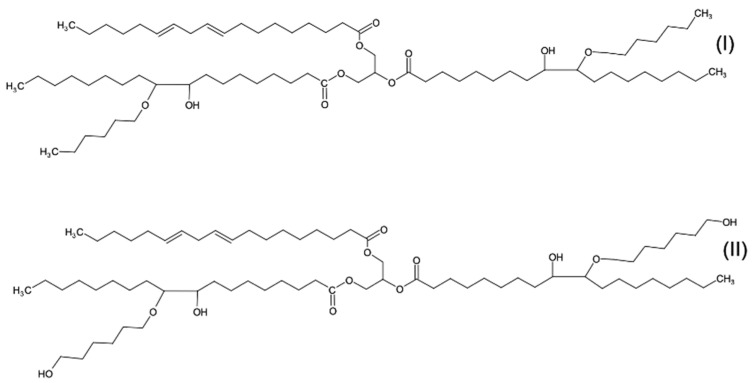
Hypothetical structure of C.HEX (**I**) and C.1.6HEX (**II**).

**Figure 2 polymers-15-03660-f002:**
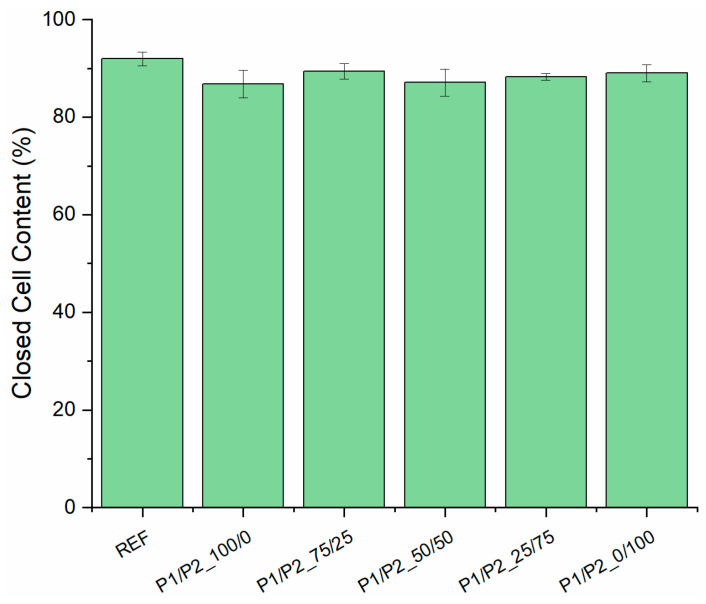
Closed-cell content in tested foams.

**Figure 3 polymers-15-03660-f003:**
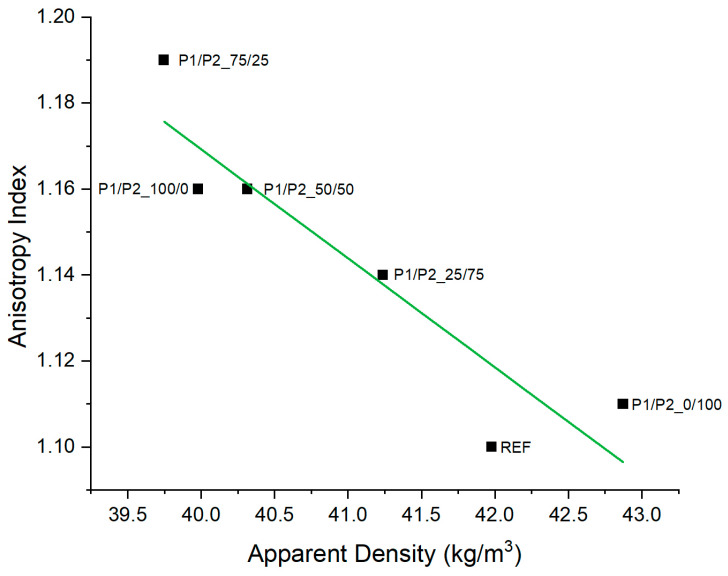
Comparison of the anisotropy index of RPURF cells in a cross-section parallel to the direction of foam growth and the apparent density of foams.

**Figure 4 polymers-15-03660-f004:**
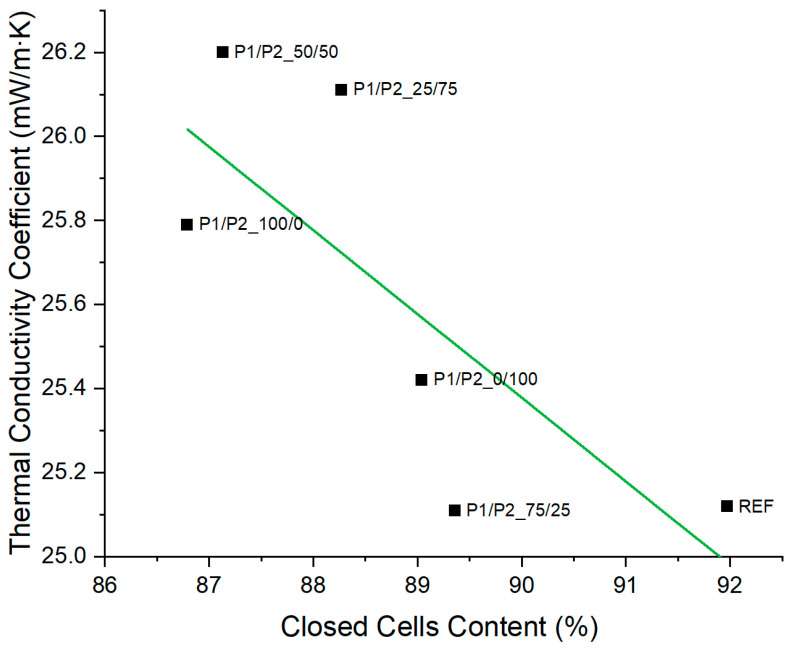
Comparison of the content of closed cells and thermal conductivity coefficient of RPURFs.

**Figure 5 polymers-15-03660-f005:**
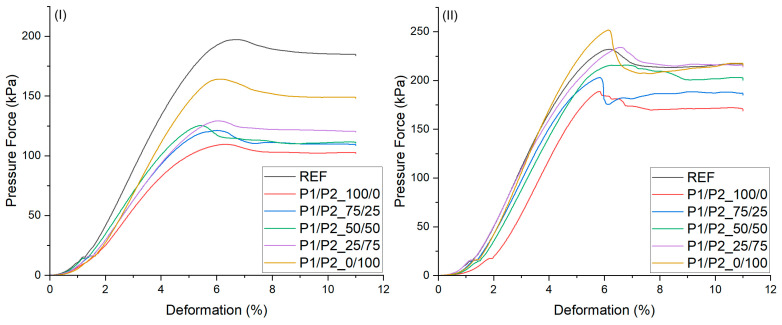
Stress–strain curves obtained from compression tests of rigid polyurethane foams, tested perpendicular (**I**) and parallel (**II**) to the direction of foam growth.

**Figure 6 polymers-15-03660-f006:**
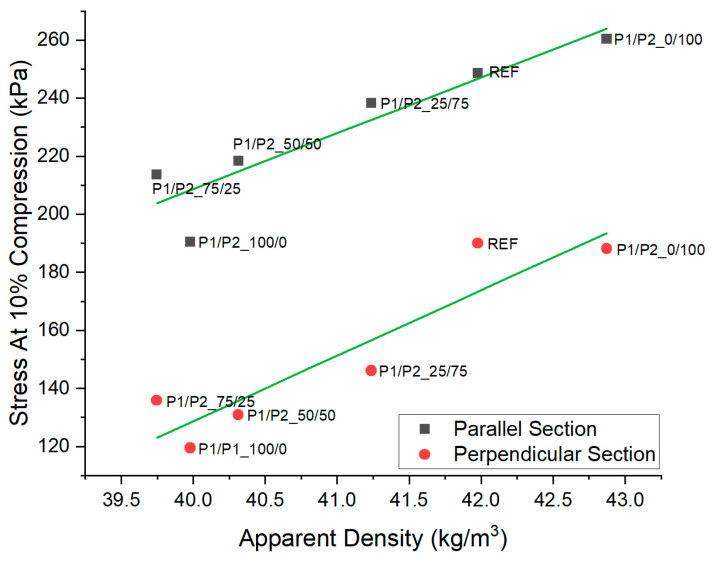
Comparison of apparent density and stress at 10% apparent compression of RPURFs.

**Figure 7 polymers-15-03660-f007:**
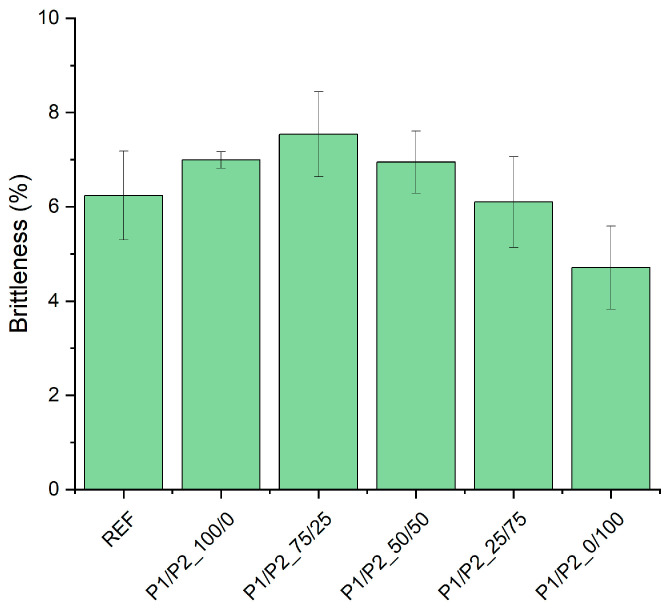
Brittleness of RPURFs.

**Figure 8 polymers-15-03660-f008:**
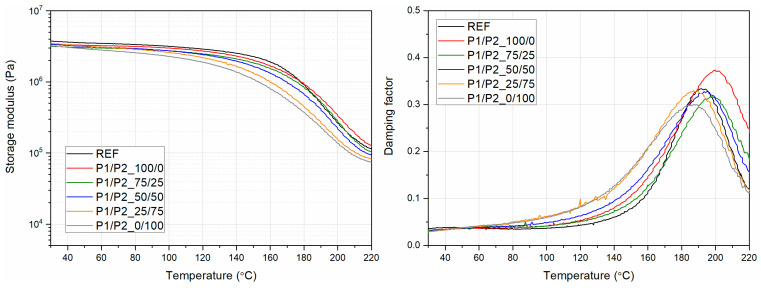
Dependence of storage modulus and damping factor on temperature for the tested RPURFs.

**Table 1 polymers-15-03660-t001:** Characteristics of petrochemical polyol and bio-polyols [31].

Properties	Petrochemical Polyol Rokopol^®^ RF 551	Bio-Polyol C.HEX	Bio-Polyol C.1.6HEX
Hydroxyl value (mgKOH⋅g^−1^)	420	104	250
Water content (%)	0.1	0.04	0.25
Viscosity (mPa⋅s)	4000	561	2050
Functionality	4.8	2.5	4.4

**Table 2 polymers-15-03660-t002:** Formulations of prepared RPURFs.

Component (g)	0% BIO	40% BIO				
	REF	P1/P2_100/0	P1/P2_75/25	P1/P2_50/50	P1/P2_25/75	P1/P2_0/100
**RF 551**	100	60	60	60	60	60
**C.HEX**	-	40	30	20	10	-
**C.1.6HEX**	-	-	10	20	30	40
**Catalyst Polycat 5**	0.50					
**Surfactant** **L-6915**	1.50					
**Water**	3.40	3.40	3.36	3.32	3.28	3.25
**PMDI**	169.6	136.0	139.8	143.7	147.5	151.6

**Table 3 polymers-15-03660-t003:** Parameters of the cellular structure of obtained foams.

Foam Symbol		Anisotropy Index	Average Cell Cross-Sectional Area 10^2^ (mm^2^)
**REF**	Parallel	1.10 ± 0.051	0.97 ± 0.209
	Perpendicular	0.98 ± 0.080	0.72 ± 0.189
**P1/P2_100/0**	Parallel	1.16 ± 0.048	0.84 ± 0.142
	Perpendicular	0.95 ± 0.025	0.58 ± 0.043
**P1/P2_75/25**	Parallel	1.19 ± 0.068	0.83 ± 0.061
	Perpendicular	0.96 ± 0.082	0.55 ± 0.086
**P1/P2_50/50**	Parallel	1.16 ± 0.079	0.87 ± 0.155
	Perpendicular	0.98 ± 0.055	0.54 ± 0.124
**P1/P2_25/75**	Parallel	1.14 ± 0.039	0.94 ± 0.057
	Perpendicular	0.94 ± 0.021	0.61 ± 0.081
**P1/P2_0/100**	Parallel	1.11 ± 0.112	1.02 ± 0.401
	Perpendicular	0.98 ± 0.063	0.63 ± 0.095

**Table 4 polymers-15-03660-t004:** Microscopic photos of tested foams.

Foam Symbol	Direction	
	Parallel	Perpendicular
**REF**	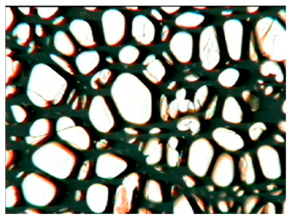	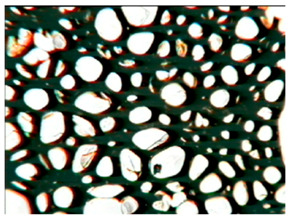
**P1/P2_100/0**	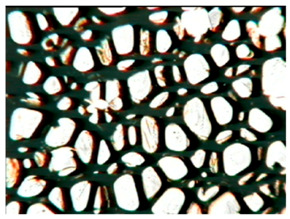	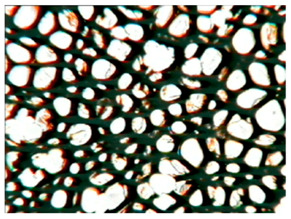
**P1/P2_75/25**	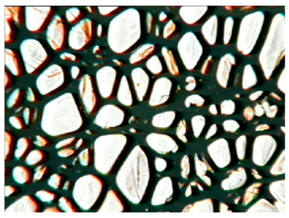	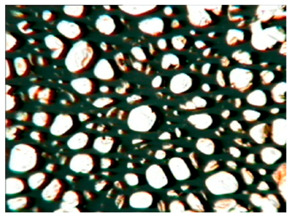
**P1/P2_50/50**	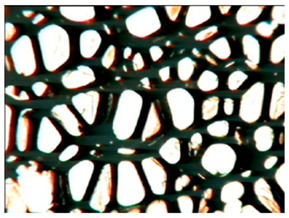	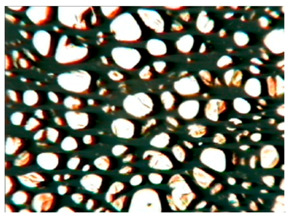
**P1/P2_25/75**	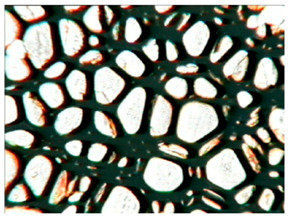	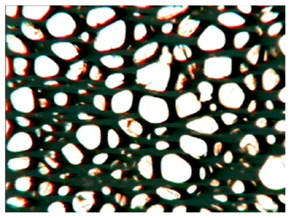
**P1/P2_0/100**	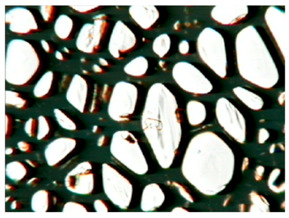	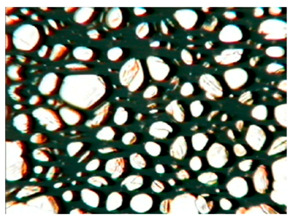
		0.5 mm

**Table 5 polymers-15-03660-t005:** Apparent density and thermal conductivity coefficient of foams.

Foam Symbol	Apparent Density (kg⋅m^−3^)	Thermal Conductivity Coefficient (mW⋅(m⋅K)^−1^)
**REF**	42.0 ± 0.25	25.1 ± 0.26
**P1/P2_100/0**	40.0 ± 0.01	25.8 ± 0.37
**P1/P2_75/25**	39.7 ± 0.38	25.1 ± 0.49
**P1/P2_50/50**	40.3 ± 0.36	26.2 ± 0.35
**P1/P2_25/75**	41.2 ± 1.12	26.1 ± 0.14
**P1/P2_0/100**	42.9 ± 0.05	25.4 ± 0.60

**Table 6 polymers-15-03660-t006:** Compression stress of RPURFs at 10% deformation.

Foam Symbol	Compression Stress at 10% Deformation (kPa)	
	Parallel	Perpendicular
**REF**	248.7 ± 17.24	190.1 ± 13.60
**P1/P2_100/0**	190.5 ± 18.17	119.5 ± 10.46
**P1/P2_75/25**	213.8 ± 11.41	135.9 ± 12.93
**P1/P2_50/50**	218.4 ± 14.15	130.9 ± 16.19
**P1/P2_25/75**	238.3 ± 10.41	146.1 ± 18.38
**P1/P2_0/100**	260.4 ± 8.88	188.1 ± 19.85

**Table 7 polymers-15-03660-t007:** Changes in linear dimensions of tested foams.

Foam Symbol	Temperature: −25 °C			Temperature: 70 °C Humidity: 90%		
	Length (%)	Width (%)	Thickness (%)	Length (%)	Width (%)	Thickness (%)
**REF**	0.10 ± 0.117	0.09 ± 0.217	−0.31 ± 0.819	0.55 ± 0.085	0.49 ± 0.102	0.43 ± 0.643
**P1/P2_100/0**	0.71 ± 0.376	0.55 ± 0.341	0.01 ± 0.565	0.56 ± 0.365	0.43 ± 0.313	−0.37 ± 0.690
**P1/P2_75/25**	0.38 ± 0.246	0.31 ± 0.126	−0.11 ± 0.580	0.64 ± 0.233	0.57 ± 0.257	−0.23 ± 0.560
**P1/P2_50/50**	0.36 ± 0.117	0.34 ± 0.188	0.02 ± 0.469	0.58 ± 0.249	0.54 ± 0.296	−0.47 ± 0.947
**P1/P2_25/75**	0.30 ± 0.188	0.35 ± 0.148	0.10 ± 0.487	0.67 ± 0.315	0.58 ± 0.313	−0.46 ± 0.687
**P1/P2_0/100**	0.21 ± 0.162	0.39 ± 0.125	0.72 ± 0.600	0.43 ± 0.172	0.54 ± 0.212	−0.69 ± 0.684

## Data Availability

Not applicable.

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
