# Peer review of "Effect of Selected Bio-Components on the Cell Structure and Properties of Rigid Polyurethane Foams"

_polymers, 2023, doi:10.3390/polym15183660_

Round 1

Reviewer 1 Report

The paper presents the results of research on the effect of new bio-polyols from rapeseed oil on the properties of the RPURF and the cellular structure and the physical and mechanical properties of the PUR composition was study in detail. This work made it possible to produce environmentally friendly thermal insulation materials in the form of RPURF with satisfactory physical and mechanical properties. But it still needs some minor revise before published.

1.       The instrument models used in the study are not listed.

2.       Fig. 2-4 The presentation of data makes people feel strange. Additionally, such data should have error bars.

3.       Conclusions part. Should include more date.

4.       The author should provide a detailed explanation of the reasons for this result in the conclusion section.

5.       Extensive editing of English language required

Extensive editing of English language required

Author Response

Dear Reviewer,

Thank you for your review and comments on the manuscript. Below are the responses to comments.

„The paper presents the results of research on the effect of new bio-polyols from rapeseed oil on the properties of the RPURF and the cellular structure and the physical and mechanical properties of the PUR composition was study in detail. This work made it possible to produce environmentally friendly thermal insulation materials in the form of RPURF with satisfactory physical and mechanical properties. But it still needs some minor revise before published.

  1. The instrument models used in the study are not listed.

The information about instrument models has been added if it was possible.

  1. Fig. 2-4 The presentation of data makes people feel strange. Additionally, such data should have error bars.

      The data presented in the mentioned figures show a tendency in the properties changes depending on the content of bio-polyols mixture. The standard deviations for the properties are given in the tables 3, 5 and 6, and in Fig. 2 (in revised version).

  1. Conclusions part. Should include more date.

      This part has been corrected.

  1. The author should provide a detailed explanation of the reasons for this result in the conclusion section.

      This part has been corrected.

  1. Extensive editing of English language required.

      The paper has been corrected.

Kind regards,

Michał Kucała

Reviewer 2 Report

1)       the ABSTRACT is not clear in defining the novelty of the paper and should be rewritten.

2)     Please provide an introduction that highlights the importance of understanding thermal conductivity and convection heat transfer in high porosity foams. Discuss the significance of studying foams with a lattice structure and their potential applications in various industries, such as thermal insulation, heat exchangers, or energy storage systems.

3)      Clarify the specific objective of the study and how it addresses the current research gaps.

4)     In the introduction section mention the variation with temperature of the thermophysical properties of polyurethane foams. Please check this reference and report in detail these findings: https://doi.org/10.1016/j.conbuildmat.2023.131980

5)     Information about the glass transition temperature, decomposition temperature should also be reported.

6)     The fact that increasing density led to an improvement in the foam mechanical properties is expected. The authors should improve the discussion about the micro-mechanisms which lead to this behaviour. Please check the book of Lorna Gibson

7)     Please include the stress vs strain curves obtained from the compression tests.

8)     How the results of the DMA tests compared with those available in the literature?

9)     Comment on any notable differences in thermal conductivity and convection heat transfer between the two foam materials. Discuss the implications of these differences for their respective applications and any suggestions for optimizing their performance in specific thermal management scenarios.

10)   In the conclusions, the authors should explain the significance and shortcomings of the research work, instead of repeating the results obtained before.

Author Response

Dear Reviewer,

Thank you for your review and comments on the manuscript. Below are the responses to comments.

1)  The ABSTRACT is not clear in defining the novelty of the paper and should be rewritten.

      It has been corrected.

2)     Please provide an introduction that highlights the importance of understanding thermal conductivity and convection heat transfer in high porosity foams. Discuss the significance of studying foams with a lattice structure and their potential applications in various industries, such as thermal insulation, heat exchangers, or energy storage systems.

      Introduction has been completed and corrected.

3)     Clarify the specific objective of the study and how it addresses the current research gaps. 

      The specific objective of the study has been clarified.

4)     In the introduction section mention the variation with temperature of the thermophysical properties of polyurethane foams. Please check this reference and report in detail these findings: https://doi.org/10.1016/j.conbuildmat.2023.131980   

      Introduction has been completed.

5)     Information about the glass transition temperature and decomposition temperature should also be reported.

The additional information about characteristic temperatures obtained by DMA has been included in the revised manuscript version.

6)     The fact that increasing density led to an improvement in the foam mechanical properties is expected. The authors should improve the discussion about the micro-mechanisms which lead to this behaviour. Please check the book of Lorna Gibson M.

      The introduction refers to the works in which the issue of the dependence of density changes on the mechanical properties of foams was analyzed. Additionally, the effects of temperature and apparent density on mechanical properties of polyurethane foams was discussed.

7)     Please include the stress vs strain curves obtained from the compression tests. M.

      The curves of the stress vs strain are in the Fig. 5 (in revised version).

8)     How the results of the DMA tests compared with those available in the literature?

A rheometer equipped with a system was used to analyze the properties of thermomechanical samples by DMTA method, fixed and deformed in the torsion mode to perform the measurements. Considering the different devices and the heat transfer methods resulting from them, resulting from the inertia of the chamber or the heating efficiency, the deformation geometries (compression, bending, and in the case under consideration, torsion), it is not reasonable to compare values such as Tg made by other researchers with those obtained. The work of Hagen et al. [10.1016/0142-9418(94)90020-5] showed a significant influence of the apparatus used on the recorded values describing changes in thermomechanical properties, including Tg. The principle, however, is the analysis of changes and dependencies between samples resulting from changes in material composition. We thank the Reviewer for this comment; relevant literature references have been cited in the text of the revised manuscript.

9)     Comment on any notable differences in thermal conductivity and convection heat transfer between the two foam materials. Discuss the implications of these differences for their respective applications and any suggestions for optimizing their performance in specific thermal management scenarios.

      The discuss on the heat transfer has been completed.

10  In the conclusions, the authors should explain the significance and shortcomings of the research work, instead of repeating the results obtained before.”

This part has been corrected. The conclusions indicate the motivation to perform the research and summarize the most important results, pointing to the role of bio-polyol functionality.

Kind regards,

Michał Kucała

Reviewer 3 Report

The paper entitled “Effect of selected bio-components on the cell structure and properties of rigid polyurethane foams” focus on the effect of new bio-polyols from rapeseed oil on the properties. Polyurethane foams (PPU) are widely used as insulating materials. The use of polyurethane foam materials for thermal insulation makes it possible to reduce heat losses many times over, heat power engineering and other industries.

Comments:

1. There is a lot of self-citation, it should not exceed than 3 times only.

2. In your work you use Bio-polyol C.HEX Bio-polyol C.1.6HEX you need to give the chemical structures of these compounds.

3. Introductory part completely rewritten.

4. Table 4. Missing measurement scales

5. Have you recycled these polyurethane foams? How will the addition of rapeseed oil affect recycling?

6. What are the advantages of rapeseed oil over other additives?

As a result, I will recommend the publication of this manuscript after major revision.  

Author Response

Dear Reviewer,

Thank you for your review and comments on the manuscript. Below are the responses to comments.

“The paper entitled “Effect of selected bio-components on the cell structure and properties of rigid polyurethane foams” focus on the effect of new bio-polyols from rapeseed oil on the properties. Polyurethane foams (PPU) are widely used as insulating materials. The use of polyurethane foam materials for thermal insulation makes it possible to reduce heat losses many times over, heat power engineering and other industries.

Comments:

  1. There is a lot of self-citation, it should not exceed than 3 times only.

It has been corrected.

  1. In your work you use Bio-polyol C.HEX Bio-polyol C.1.6HEX you need to give the chemical structures of these compounds.

The chemical structures of the bio-polyols have been added.

  1. Introductory part completely rewritten.

This part has been corrected.

  1. Table 4. Missing measurement scales

The measurement scale has been added.

  1. Have you recycled these polyurethane foams? How will the addition of rapeseed oil affect recycling?

The introduction of bio-polyols from rapeseed oil affects the process of chemolysis. Currently, the research team is implementing a project to analyze the impact of the type and content of bio-polyol on the properties of recyclates. Preliminary studies carried out as part of the project have shown beneficial effects, among others on the viscosity of recyclates. With the increase in the content of biopolyols with a specific chemical structure, the viscosity of the recyclate decreases, which is beneficial from the application point of view of such components.

  1. What are the advantages of rapeseed oil over other additives?

Vegetable oils, due to the content of unsaturated and ester bonds, make it possible to obtain various hydroxyl derivatives. Such derivatives are capable of reacting with isocyanates and forming a polymer matrix. Previous studies indicate that even 100% of petrochemical polyols in some foam systems can be replaced with bio-polyols from vegetable oils, including used frying oils.

Kind regards,

Michał Kucała

Round 2

Reviewer 2 Report

Although the authors did an effort to improve the quality of the paper by addressing most of the comments raised by the reviewers, there are still several aspect that need to be clarified/discussed in more details.

1)      the variation with temperature of the thermophysical properties of polyurethane foams is not clear in the introduction. Please report the findings obtained in the study suggested in the previous round to provide a completer overview to the reader.

2)      Authors reported the dynamic compressive response of PUR foam, but the static compressive/shear/tensile response of this material at elevated temperatures is missing. Please add this information

3)      Proper Information about the glass transition temperature and decomposition temperature of PUR foam is still missing in the introduction!

4)      The reply to remark response 6 is not clear.

5)      Concerning the compressive stress vs strain curves; how did you measured strains? In addition how do you explain the different post-yielding softening among the specimens tested?

Author Response

Dear Reviewer,

Thank you for your review and comments on the manuscript. Below are the responses to comments.

1) The variation with temperature of the thermophysical properties of polyurethane foams is not clear in the introduction. Please report the findings obtained in the study suggested in the previous round to provide a completer overview to the reader.

Information on the effect of temperature on selected thermophysical properties of polyurethane foams has been added.

2)  Authors reported the dynamic compressive response of PUR foam, but the static compressive/shear/tensile response of this material at elevated temperatures is missing. Please add this information

The static mechanical properties at elevated temperatures were not investigated.

3) Proper Information about the glass transition temperature and decomposition temperature of PUR foam is still missing in the introduction!

Information about the glass transition temperature and decomposition temperature of PUR foam has been added in the introduction.

4) The reply to remark response 6 is not clear.

This part has been completed on the base of the book of Lorna Gibson

5) Concerning the compressive stress vs strain curves; how did you measured strains? In addition how do you explain the different post-yielding softening among the specimens tested?

The stain was calculated by the applied ZwickRoell testing machine, as the sample deformation in the direction of the applied force divided by the initial high of the sample. The movement of the compression platen is measured by the crosshead travel of the testing machine.

Kind regards,

Michał Kucała

Reviewer 3 Report

Corrections have been made for better understanding. article can be accepted in this form

Author Response

Dear Reviewer,

Thank you for your review and comments on the manuscript.

Kind regards,

Michał Kucała

Round 3

Reviewer 2 Report

The paper is significantly improved following the review process. The authors responded to all the reviewers' comments, therefore, I recommend that the paper be published in the current form. Of course, the final decision belongs to the Editor.